# Biomimetic Remineralization of Artificial Caries Lesions with a Calcium Coacervate, Its Components and Self-Assembling Peptide P_11_-4 In Vitro

**DOI:** 10.3390/bioengineering11050465

**Published:** 2024-05-08

**Authors:** Basel Kharbot, Haitham Askar, Dominik Gruber, Sebastian Paris

**Affiliations:** 1Department of Operative, Preventive and Pediatric Dentistry, Charité—Universitätsmedizin Berlin, 14917 Berlin, Germany; 2Physical Chemistry, Department of Chemistry, University of Konstanz, 78464 Konstanz, Germany

**Keywords:** bio-inspired materials, crystallization, dental caries, tooth remineralization

## Abstract

The application of calcium coacervates (CCs) may hold promise for dental hard tissue remineralization. The aim of this study was to evaluate the effect of the infiltration of artificial enamel lesions with a CC and its single components including polyacrylic acid (PAA) compared to that of the self-assembling peptide P_11_-4 in a pH-cycling (pHC) model. Enamel specimens were prepared from bovine incisors, partly varnished, and stored in demineralizing solution (DS; pH 4.95; 17 d) to create two enamel lesions per sample. The specimens were randomly allocated to six groups (n = 15). While one lesion per specimen served as the no-treatment control (NTC), another lesion (treatment, T) was etched (H_3_PO_4_, 5 s), air-dried and subsequently infiltrated for 10 min with either a CC (10 mg/mL PAA, 50 mM CaCl_2_ (Ca) and 1 M K_2_HPO_4_ (PO_4_)) (groups CC and CC + DS) or its components PAA, Ca or PO_4_. As a commercial control, the self-assembling peptide P_11_-4 (Curodont^TM^ Repair, Credentis, Switzerland) was tested. The specimens were cut perpendicularly to the lesions, with half serving as the baseline (BL) while the other half was exposed to either a demineralization solution for 20 d (pH 4.95; group CC + DS) or pHC for 28 d (pH 4.95, 3 h; pH 7, 21 h; all five of the other groups). The difference in integrated mineral loss between the lesions at BL and after the DS or pHC, respectively, was analyzed using transversal microradiography (ΔΔZ = ΔZ_pHC_ − ΔZ_baseline_). Compared to the NTC, the mineral gain in the T group was significantly higher in the CC + DS, CC and PAA (*p* < 0.05, Wilcoxon). In all of the other groups, no significant differences between treated and untreated lesions were detected (*p* > 0.05). Infiltration with the CC and PAA resulted in a consistent mineral gain throughout the lesion body. The CC as well as its component PAA alone promoted the remineralization of artificial caries lesions in the tested pHC model. Infiltration with PAA further resulted in mineral gain in deeper areas of the lesion body.

## 1. Introduction

The management of early caries lesions includes non- as well as micro-invasive treatment options. Non-invasive strategies aim to prevent lesion formation or progression through dietary modifications, biofilm control and remineralization. These approaches directly intervene in the caries process and have limited negative effects but often depend on patient adherence. Micro-invasive treatment strategies aim to arrest non-cavitated caries lesions via the obstruction of lesion pores, e.g., by infiltration with a light-curing resin [1,2,3,4]. After the removal of the hypermineralized surface layer, a low-viscosity resin is applied to infiltrate the porous lesion body by capillary forces [5,6]. In this study, we propose combining the advantages of non- and micro-invasive concepts by aiming to remineralize artificial carious lesions in bovine enamel specimens with different agents using an infiltration method. By delivering remineralizing agents rapidly to the lesion body driven by capillary forces, a common hurdle in preservative dentistry is addressed. Remineralization in the presence of fluorides commonly takes place on the surfaces of carious lesions [7,8]. Hence, the lesion body showing the primary mineral loss stays demineralized. Also, the availability of calcium and phosphate ions is usually a limiting factor during topical fluoride application [9]. Several approaches aiming at a biomimetic enamel regeneration to overcome these hurdles include the use of remineralizing agents, such as casein phosphopeptide and amorphous calcium–phosphate (CPP-ACP), CCP-ACP with functional fluoride CPP-ACP(F), nano-hydroxyapatite, or the self-assembling peptide P_11_-4 [10,11,12,13,14,15,16,17]. P_11_-4 is a rationally designed, oligomeric ß-sheet-forming peptide with the ability to self-assemble into three-dimensional scaffolds [18]. Its application in the dental field relies on the affinity for Ca^2+^ ions to act as a nucleation site for de novo hydroxyapatite formation [19,20]. Clinically, fluorides remain the cornerstone for early caries management with the highest level of supporting evidence, although they seem to be the most effective when used in combination with other treatment measures, such as resin infiltration, especially in high-risk patients [21,22,23,24,25]. In previous studies, we attempted to initiate remineralization processes in enamel lesions exploring bio-inspired possibilities of non-classical crystallization (NCC) pathways through liquid precursors [26].

NCC is used as a term to summarize crystallization processes that, in contrast to classical crystallization processes, do not include atom, ion or molecular building units but have larger building units, such as liquid droplets, that function as amorphous precursors and self-assemble to form superstructures [27,28,29]. Moreover, NCC can be initiated independently from free crystal surfaces and can facilitate mineral growth much faster than the classical layer-by-layer crystallization process [30]. One of the various known NCC pathways observed in biomineralization includes polymer-induced liquid precursor (PILP) phases. PILPs are amorphous mineral precursors stabilized by charged polymers, e.g., polyacrylic acid (PAA) [31]. Polyacrylic acid is an anionic polymer with pendent carboxyl side chains and is soluble in aqueous media at a neutral pH [32]. Several PILP applications, including anionic polymers such as PAA, have been reported in the biomedical field to promote biomimetic hard tissue regeneration, including bone tissue and dental hard tissues [32,33,34,35,36,37]. In dentin, Tay et al. used PAA as a calcium phosphate- and collagen-binding matrix protein analogue to achieve intra- and interfibrillar remineralization via the formation of metastable amorphous calcium phosphate nanoprecursors [33,38]. We recently presented PAA as a polymer interacting with Ca^2+^ with its carboxylic groups to form (PAA–Ca^2+^) coacervate droplets which are delivered to the porous lesion body using capillary forces and possibly form de novo hydroxyapatite (Prause et al., in review). To overcome the physiological limitation of restricted phosphate availability, lesions were initially infiltrated with phosphate ions serving as a depot, followed by CC infiltration, allowing for the formation of calcium phosphate nanocrystals in a stabilizing PAA matrix.

Coacervation generally describes a phase separation of partly soluble colloids in a solvent, resulting in a dense and droplet-rich coacervate phase and a dilute phase [39]. In this study, complex coacervation occurs based on the electrostatic interaction of polyacrylic acid and calcium ions in an aqueous solution, forming PAA–Ca^2+^ droplets [40,41,42,43,44]. The infiltration of enamel caries lesions with a calcium coacervate emulsion (CC) in our previous studies did not yield significant, immediate remineralization, but rather showed protective effects against further demineralization with characteristic mineralized bands within the lesion bodies [26]. However, the question of whether this effect was achieved by the CC or one of its components could not be definitively answered. We suggest that the residues of the infiltrated PAA in the caries lesions might play a role in the changed dynamics of demineralization. Within this context, the aim of this study was to investigate the effects of the infiltration of artificial enamel caries lesions with a CC as well as its single components (including PAA) compared to that of the commercially available product P_11_-4 (Curodont^TM^ Repair) in a pH-cycling (pHC) model. Our null hypothesis was that there is no significant difference in the integrated mineral loss of caries lesions after treatment with the tested agents in a pHC model.

## 2. Materials and Methods

### 2.1. Study Design

An a priori power analysis was conducted using Sample Size Calculator (Version 1.061) for sample size estimation. Based on a previous study [26], the effect size was considered to be small using Cohen’s (1988) criteria. With a significance level of α = 0.05 and power = 0.80, the minimum sample size required was calculated to be 15 assuming a drop-off rate of 10%.

For this study, 90 bovine enamel specimens were prepared, and two artificial caries lesions in each specimen were created (Figure 1). Specimens were randomly allocated to 6 groups. One lesion per specimen served as a non-treatment control (NTC) while the other lesion (T) was treated with one of five experimental materials: CC (CC and CC + DS), PAA-Na (PAA), CaCl_2_·2 H_2_O (Ca), K_2_HPO_4_ (PO_4_) or self-assembling peptide P_11_-4 (P_11_-4). Specimens were subsequently cut perpendicularly to the lesions, with one half serving as baseline control (BL) while the other half (effect, E) underwent pHC. To test the reproducibility of a previous study in one group (CC + DS), the E-halves of CC-treated lesions were exposed to a demineralizing solution instead of pHC [26]. The change in integrated mineral loss by different treatment modalities was analyzed using transversal microradiography (TMR).

### 2.2. Specimen Preparation

Permanent bovine incisors were obtained and stored in chloramine-T solution (0.5%). The roots of the teeth were separated, and the crowns prepared and ground flat to obtain standardized enamel specimens (7 × 4 × 3 mm^3^; n = 93) using a band saw and permanent water cooling (EXAKT 300cl, EXAKT, Norderstedt, Germany). The enamel blocks were embedded in acrylic resin (Technovit 4071; Heraeus Kulzer, Hanau, Germany) and polished (with grain sizes of 1200, 2500 and 4000; Struers, Willich, Germany).

On each of the 90 specimens, two areas were covered with acid-resistant nail varnish to obtain a sound enamel area (S) and two separate, unprotected enamel windows for creating the artificial caries lesions. Specimens were stored in a demineralization solution [45] containing 50 mM acetic acid, 3 mM CaCl_2_∙2H_2_O, 3 mM KH_2_PO_4_ and 6 µL Methyl-hydroxy-diphosphonate (pH of 4.95; 37 °C) with the enamel surfaces facing up for 22 d. The demineralization solution was prepared to cover the specimens completely. The pH level was monitored daily and adjusted with 10% hydrochloric acid and 10 M potassium hydroxide solution, if necessary. After demineralization, the samples were stored in a humid chamber at 3 °C until further utilization.

### 2.3. Specimen Treatment

One lesion (T) of each specimen in the CC + DS, CC, PAA, Ca and PO_4_ groups was etched using 37% phosphoric acid gel (Total Etch; Vivadent, Schaan, Liechtenstein) for 5 s to remove the pseudo-intact surface layer. The acid gel was rinsed off for 30 s, and the surfaces were thoroughly air-dried for 10 s in preparation for the infiltration treatment.

The CC + DS and CC groups were both infiltrated with coacervate emulsion following the same procedure as follows: 1 M K_2_HPO_4_ was applied for 3 min to allow for the loading of the lesions with phosphate ions and subsequently air-dried for 10 s. The coacervate emulsion was prepared by transferring 500 µL polyacrylic acid sodium salt (PAA-Na, 10 mg/mL, Mw = 15 kDa, pH = 9; Sigma-Aldrich, St. Louis, MO, USA) into Eppendorf tubes and adding 450 µL calcium chloride dihydrate (CaCl_2_·2 H_2_O, 50 mM, pH = 6; Carl Roth, Karlsruhe, Germany) under continuous stirring at appr. 23 °C. The tubes were properly shaken for 3 s during which separation and formation of coacervate emulsions were macroscopically visible due to their milky appearance. The solutions were immediately applied to the dried lesions and left for 10 min. Excess was gently removed, and the specimens were stored in a humid chamber at room temperature.

In PAA, Ca and PO_4_ groups, the treatment solutions were directly applied after the etching process and left on the lesion surfaces for a duration of 10 min. In P_11_-4, Curodont^TM^ Repair (Credentis AG, Windisch, Switzerland) was applied according to the manufacturer’s instructions with exception of a slightly modified etching step and prolonged infiltration time (10 min instead of 5 min). The surface was cleansed using a small cotton pallet soaked with 3% hypochloride solution for 20 s. After rinsing and air-drying the lesions, the surface area was etched using 37% phosphoric acid for 5 s (the instructions recommend 20 s), since a relatively short etching time has been proven to be sufficient for artificial lesions and longer etching time would have destroyed the artificial lesions [5,46]. The etching gel was rinsed for 30 s, lesions were gently dried and the Curodont^TM^ Repair agent was applied and left on the specimens for a duration of 10 min before removing excess.

After treatment with the various solutions/emulsions, the specimens were cut perpendicularly to the lesions, yielding two separate halves for each specimen. The sections serving as BL were coated with nail polish to protect the lesion surfaces until further processing. The sections serving as E were coated on the cut surfaces to avoid penetration of fluids into the enamel before exposition to a demineralization phase or pHC, respectively.

### 2.4. Demineralization/pH Cycling

Specimens of CC + DS group went through a second demineralization phase for 20 d (at a pH of 4.95; 37 °C) following the previous protocol for creating caries lesions. All other groups underwent pHC for 28 d to simulate oral fluctuation patterns (Table 1). The pHC consisted of demineralization periods of 21 h and remineralization periods of 3 h and was performed automatically by setting up a programmable pumping machine system (Persitaltic pumps P4, Seko, Rufina, Italy; EG-PM2 programmable power outlet strip, Energenie, Almere, The Netherlands). For demineralization, the previously described solution was used again (at a pH of 4.95; 37 °C). The remineralization solution contained 1.5 mM CaCl_2_∙2H_2_O, 0.9 mM KH_2_PO_4_, 130 mM KCl, 20 mM hydroxyethylpiperazine-N’-2-ethanesulfonic acid (HEPES) buffer, 3 mM NaN_3_ and 5.26 mM F^−^ (pH 7; 37 °C) [45]. Between period changes, short rinsing cycles with water took place to prevent inconsistent pH levels due to solution residues.

### 2.5. Transversal Microradiographic Analysis

Transversal microradiography (TMR) was used for quantitative analysis of mineral loss (ΔZ) and lesion depth (LD) of the lesions. The specimens were attached to an object holder to prepare and polish thin sections (Band Saw Exact 300cl; Exakt Apparatebau, Norderstedt, Germany; grain sizes of 1200, 2500 and 4000) to a final thickness of 100 µm ± 10 µm. The thin sections were X-rayed with the inclusion of an aluminium step-wedge for calibration (PW2213/20 tube, Panalytical, Kassel, Germany; PW 1730/10 generator, Philips, Eindhoven, The Netherlands) at 20 kV and 10 mA (exposure time of 10 s). The exposed films (Fine 71337, Fujifilm, Tokyo, Japan) were developed according to the manufacturer’s instructions and under standardized conditions. Digital analyzation of the microradiographs with respect to integrated ΔZ and LD was performed using a CCD video camera module (XC77CE, Sony, Tokyo, Japan) attached to a transmitted light microscope (Axioskop2 60318, Zeiss, Oberkochen, Germany) with dedicated software (TMR for Windows 2.0.27.2, Inspektor Research, Amsterdam, The Netherlands).

### 2.6. Statistical Analysis

SPSS (SPSS 27.0.0.0, SPSS, Inc., Chicago, IL, USA) was used for data analysis. To calculate differences in mineral loss of the lesions before and after pHC or DS, respectively, values of BL were subtracted from E (ΔΔZ = ΔZ_E_ − ΔZ_BL_) for both T and NTC lesions.

The Kolmogorov–Smirnov test was used to assess the values for integrated mineral loss for normal distribution. Since not all values were normally distributed, the Wilcoxon test was applied to identify statistically significant differences (set at *p* ≤ 0.05) within groups. Mineral distribution within the lesions was evaluated by plotting mean mineral content against the lesion depth in steps of 0.5 µm.

## 3. Results

The untreated caries lesions showed a median (Q25/Q75) mineral loss (ΔZ) of 6326 (5391/7480) vol% × µm with a median lesion depth (LD) of 187 (171/206) µm.

In the CC + DS group, the lesions infiltrated with the CC emulsion showed significantly less mineral loss after the demineralization phase than the NTCs (ΔΔZ_T_: −56 (−1397/995) vol% × µm; ΔΔZ_NTC_: 654 (30/1766) vol% × µm; *p* < 0.05, Wilcoxon). Overall, the CC-treated lesions even showed a slight mineral gain after demineralization.

Compared to the NTCs, the mineral gain of the treated lesions was significantly higher after pHC in the CC (ΔΔZ_T_: −3372 (−4380/−371) vol% × µm; ΔΔZ_NTC_: −1171 (−3722/−209) vol% × µm; *p* < 0.05, Wilcoxon) and PAA (ΔΔZ_T_: −2370 (−3780/1248) vol% × µm; ΔΔZ_NTC_: −1020 (−1588/−216) vol% × µm; *p* < 0.05, Wilcoxon) groups (Figure 2). In all of the other groups, no significant differences between the treated and untreated lesions were detected (*p* > 0.05).

Our plots of the mineral content distribution in the effect specimen halves of the CC- and PAA-treated lesions in comparison with the NTC halves showed increased mineral contents within the entire lesion body (Figure 3), whereas few differences between the NTC and treated lesions were observed in the P_11_-4-group. While in all of the other groups, pseudo-intact surface layers were present after pHC, in the PAA-treated specimens, no superficial mineral-rich layer was present but a deeper mineralization of the lesion body was observed.

## 4. Discussion

Alternative remineralization agents to bio-mimetically remineralize enamel tissue in non-cavitated, initial caries lesions have been widely investigated [9,10,11,12,13,14]. The goal remains to achieve effective mineralization within the lesion body by forming hydroxyapatite (HAP) or fluorapatite (FAP). This study investigated the potential remineralizing effects of CC, its single components and P_11_-4 on artificial caries lesions in a pHC model. We detected significant differences in integrated mineral loss in the CC and PAA groups. Hence, the null hypothesis of this study was rejected for the CC and PAA groups but confirmed for all of the other groups.

Natural enamel development is a complex process involving various enamel matrix proteins in physiologic amelogenesis to build a highly organized matrix of rods and interrod crystals. Hence, regenerative approaches aim to emulate the functions of self-assembling proteins (e.g., amelogenin) to induce remineralization processes by creating a structured scaffold to nucleate and guide the crystallization of HAP [17,18,47,48]. The rationally designed self-assembling peptide P_11_-4 (tested here in the form of a commercially available product, Curodont^TM^ Repair) has been introduced as a potential agent to mimic these functions [12,49]. The advantage of this anionic peptide over fluoride is considered to be its ability to penetrate the subsurface of the lesion, form a 3D matrix and induce mineralization through electrostatic attraction when Ca^2+^- PO_4_^3−^-ions from oral saliva diffuse into the lesion [18,50]. However, clinical studies show partially contradictive outcomes and have limited follow-up periods of 6–12 months [17,51,52,53,54,55]. Also, easily accessible white spot lesions in occlusal areas have been assessed, and (the more relevant) proximal caries lesions have not been covered by these studies. In one study on freshly erupted molars and two studies including adults, significant improvements in all outcome parameters were reported, while three different clinical trials show no preferable outcomes for P_11_-4 compared to the gold standard, fluoride [17,51,52,53,54,55]. Since P_11_-4 becomes an elastomeric nematic gel at a pH level < 7.4 (being a low-viscosity isotropic liquid under non-acidic conditions), the relatively long remineralization phases of the pHC implemented in this study might be disadvantageous for the functional mechanisms of P_11_-4 since this specific setup yielded only a non-significant effect.

Previously, we were able to validate the X-ray diffraction patterns of HAP in CC-infiltrated artificial enamel lesions (Prause et al., in review) [26]. While the limited amount of Ca^2+^ ions that can be transferred into the lesions is most probably the reason we only observed moderate immediate effects, we were able to show the protective effects of CCs against further demineralization with highly mineralized bands within the lesion bodies. We suggested that the remaining PAA in the lesions might be able to change the dynamics of demineralizing processes due to its affinity for Ca^2+^, ability to bind the dissolved ions or function as a stabilizing matrix for calcium phosphate compounds. When lesions were exposed to remineralizing conditions, however, a highly mineralized surface layer was observed [26]. Hence, our aim in the present study was to confirm our findings of the protective effect of a CC for further demineralization and to evaluate the effects of a CC as well as PAA on caries lesions under fluctuating pH conditions in a pHC model.

We applied an established protocol for the resin infiltration method, including surface layer erosion and the desiccation of the lesions to transfer the different materials into the lesion bodies using capillary forces [5,26]. In case of the CC, phosphates were infiltrated separately in a previous step to avoid a premature precipitation of the calcium phosphate and provide necessary reaction partners for the calcium coacervates forming HAP.

Within this study, the CC was able to enhance the remineralization of artificial enamel caries lesions under fluctuating pH conditions. Nevertheless, the lesions still showed a pronounced mineralization in the surface layers (Figure 3). Most likely, these results, which are similar to our earlier results (after remineralization for 20 d), stem from the pHC implemented in this study, which tends towards remineralization with short demineralization periods of 3 h/d. Since the Ca^2+^-bound PAA within the CC was transferred into the lesions in limited amounts, a regular remineralization process as observed in the other groups most probably took place within the rather long remineralization periods, while the remaining PAA within the lesion body caused mineral retention in the demineralizing periods. However, PAA showed more pronounced effects with stronger mineralization and characteristic mineralized bands within the lesion bodies (Figure 3) instead of the lesion surfaces, solidifying our thesis that PAA is the driver for the described effects in the CC groups. We presume that the less viscous PAA solution penetrates the lesion body more efficiently than the larger and more viscous PAA–Ca^2+^ droplets. Consequently, the freely available carboxyl groups of PAA most likely bind the dissolved Ca^2+^ ions more effectively and in deeper lesion areas that the CC was not able to access in significant amounts. A previous study showed that the molar mass of polyacrylic acids and their sodium salts has a significant effect on their binding properties to HAP, rising with higher molar masses and raising the question of the potential effect in this application setting since we used a PAA–Na solution of 15 kDa (with a relatively low molecular mass) [56]. Beside the molecular weight, various studies have shown that the functioning role of PAA in biomineralization processes is highly dependent on its concentration [33,57,58]. An ideal concentration of PAA of 0.5 mg/mL to stabilize ACP in a precursor stage within the remineralization process of dentin has been reported, noting that lower concentrations only resulted in superficial remineralization while higher concentrations effectively inhibited crystallization [57]. Since this study introduces a different approach without the preparation of an ACP-containing solution, it can be assumed that the concentration of PAA will have different effects in this setting. Future studies should also investigate the effects of PAA concentrations on the mineralization processes observed in this study.

The well-established experimental setup used in this study has some strengths as well as limitations. First, bovine enamel is a widely used and viable alternative for investigations into enamel caries. Bovine teeth are easily accessible in high numbers, and the nearly identical chemical composition and radiodensity compared to those of human enamel allows for reliable radiographic evaluations of de- and remineralization processes [59,60]. Further, bovine enamel offers larger surfaces for specimen preparation and shows less variability in fluoridation than human teeth and therefore react more evenly [61]. Second, the primary outcome of mineral loss is widely accepted. The pHC model that was implemented in this study has also been reported previously and accepted for the evaluation of caries development in bovine teeth [62]. However, since the observation period in this study was only 28 d within the pHC model, a relevant question of how stable the expected PAA–Ca^2+^ compounds are remains unanswered. As a first step, longer observation periods in a similar setting and with changing pH balances (e.g., longer or more frequent demineralization phases) should be considered. Further, a pHC model is unable to depict important factors of intraoral conditions leading to re- or demineralization, e.g., bacterial biofilms or saliva. Common setups also fail to consider topical differences in an oral cavity affecting the availability and exposition of relevant variables. Also, the timing of de- and remineralizing periods differs from in vivo conditions, in which demineralizing periods normally are not as fast as those in a pHC model but occur more than once within a 24 h period [63]. Within the scope of exploring novel questions or material development, however, pHC models present a viable method to mimic the intraoral dynamics of mineral loss, ensuring a lower level of variability than that of in vivo conditions, as well as a high level of scientific controllability and comparability to past and future studies [64,65].

## 5. Conclusions

Within the limitations of this in vitro study, PAA showed an ability to influence the dynamics of re- and demineralization under fluctuating as well as demineralizing pH conditions in artificial enamel caries lesions. Our findings are compatible with previous research on the effects of CCs in enamel caries lesions and suggest that infiltration with PAA can result in mineral gain within lesion bodies as well as resistance against further demineralization.

## Figures and Tables

**Figure 1 bioengineering-11-00465-f001:**
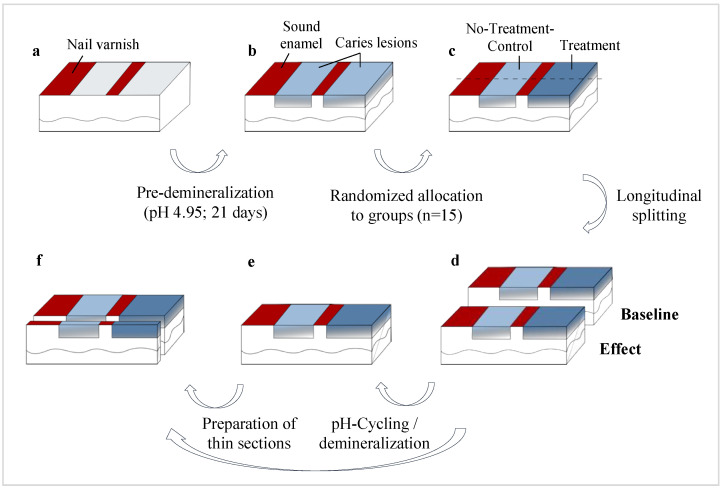
Preparation of specimens and experimental design. The specimens were polished and covered with nail varnish (red) to maintain a sound enamel area and to separate the remaining surface into two areas (**a**). Caries lesions (blue) were created (**b**). After the surface areas *Treatment* were infiltrated with the respective agents (**c**), each specimen was split longitudinally (**d**). The *Effect* halves went through a pHC or second demineralization phase, respectively (**e**), while the *Baseline* halves proceeded immediately to TMR (**f**).

**Figure 2 bioengineering-11-00465-f002:**
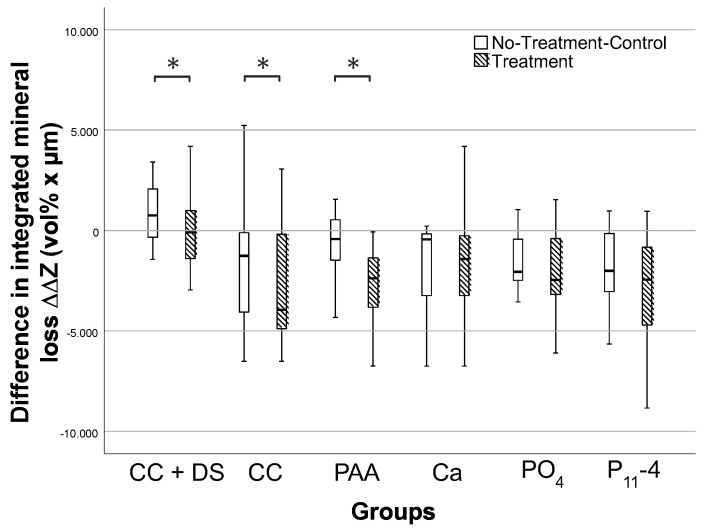
Differences in integrated mineral loss ΔΔZ (vol% × µm) in treated lesions (T; hatched boxes) and no-treatment controls (NTC; white boxes) in effect specimen halves (E) compared to baseline halves (BL). Lines: median; boxes: 25th and 75th percentiles; whiskers: minimum–maximum. Asterisks indicate significant differences in mineral loss/gain in groups CC + DS, CC and PAA.

**Figure 3 bioengineering-11-00465-f003:**
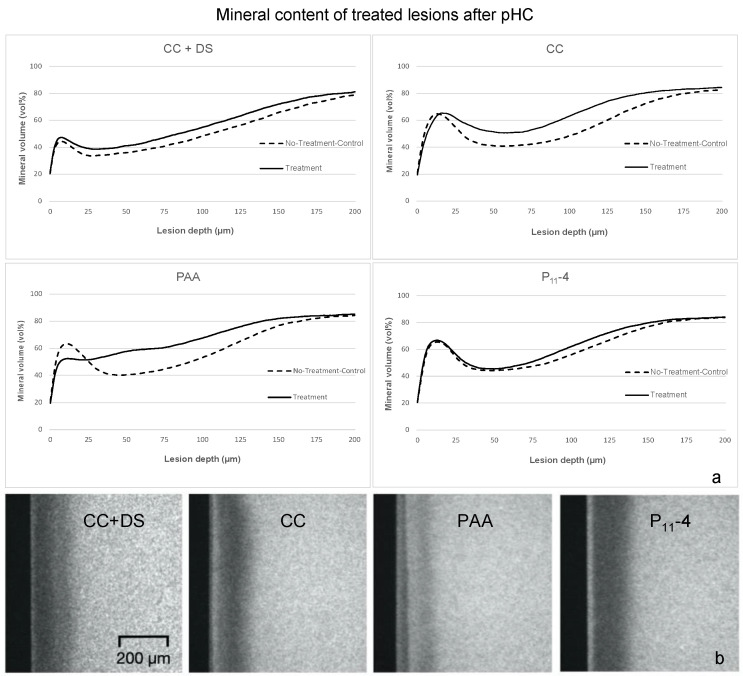
Plot of mean mineral volume throughout lesion bodies in the different treatment groups in effect specimen halves (E) (**a**) and exemplary TMR images of corresponding lesions (**b**). Meanwhile, infiltration with both CC and PAA resulted in a consistent mineral gain throughout the lesion body. However, PAA yielded a more pronounced and deeper remineralization with characteristic mineralized bands and showed no superficial mineral-rich layer.

**Table 1 bioengineering-11-00465-t001:** Treatment of experimental groups.

Group	Treatment Agent	Post-Treatment Exposition
CC	Calcium–coacervate solution	pH cycling
CC + DS	Calcium–coacervate solution	Demineralization solution
PAA	Polyacrylic acid solution	pH cycling
P_11_-4	Self-assembling peptide (Curodont^TM^ Repair)	pH cycling
PO_4_	K_2_HPO_4_	pH cycling
Ca	CaCl_2_·2 H_2_O	pH cycling

## Data Availability

Data are contained within the article.

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
