# Peer review of "Biomimetic Remineralization of Artificial Caries Lesions with a Calcium Coacervate, Its Components and Self-Assembling Peptide P11-4 In Vitro"

_bioengineering, 2024, doi:10.3390/bioengineering11050465_

Round 1

Reviewer 1 Report

Comments and Suggestions for Authors

Dear authors,

This paper is interesting but lacks scientific knowledge.

First in the abstract: please dont repeat the full name twice. Dont put abbreviation without full name. Lacks of level of siginificance. The background is not enough for readers to understand.

Keywords should be arranged according to mesh terms and alphabetical order.

Please make a null hypothesis and reject or accept it in the discussion section

Why you choose bovine incisor and explain more how did you make the deminiralization and what the difference with permanent or lacteal one.

Where is the sample size calculation?

You repeat the conclusion twice at the end of discussion section and in the discussion section.

Please make the limitations of the study.

For the results put the standard deviation with the values as quantitative not only boxplot.

Comments on the Quality of English Language

Moderate corrections

Author Response

We would like to thank the reviewer for the considerations and constructive feedback to help to improve the manuscript. Below please find our responses.

Reply to reviewer #1

1- First in the abstract: please don’t repeat the full name twice. Dont put abbreviation without full name. The background is not enough for readers to understand.

The full spellings and corresponding abbreviations of the terms concerned were checked and corrected.

The background of the study is only roughly presented in one sentence due to the word limit of the abstract according to the instructions for authors. The background of the study concept is laid out in detail in the introduction part for full understanding of context and conceptualization. Nevertheless, if more background is required in the abstract we are pleased to add more context after information.

2- Keywords should be arranged according to mesh terms and alphabetical order.

Keywords are now according to MeSH terms and ordered alphabetically:

Keywords: Bio-inspired Materials, Crystallization, Dental Caries, Tooth Remineralization

3- Please make a null hypothesis and reject or accept it in the discussion section

A null hypothesis has been stated in the Introduction and addressed again in the Discussion:

Introduction (last paragraph)

“The null hypothesis stated that there is no significant difference in integrated mineral loss of caries lesions after treatment with the tested agents in a pHC model.”

Discussion (1st paragraph)

“This study investigated the potential remineralizing effects of CC, its single components as well as P11-4 on artificial caries lesions in a pHC model. We detected significant differences in integrated mineral loss in groups CC and PAA. Hence, the null hypothesis of the study was rejected for CC and PAA but confirmed for all other groups.”

4- Why you choose bovine incisor and explain more how did you make the deminiralization and what the difference with permanent or lacteal one.

We added an explanation for the usage of bovine teeth in this study:

Discussion (last paragraph)

First, bovine enamel offers a widely used and viable alternative for the investigation of enamel caries. Bovine teeth are easily accessible in high numbers and the nearly identical chemical composition and radiodensity compared to human enamel allows reliable radiographic evaluations of de- and remineralization processes [59,60]. Further, bovine enamel offers larger surfaces for specimen preparation and shows less variability in fluoridation than human teeth and therefore react more evenly [61]."

We further described the demineralization process of the specimens in more detail:

Materials and Methods (Subjection 2.2. Specimen preparation (2nd paragraph)):

„On each of the 90 specimen, two areas were covered with acid-resistant nail varnish to obtain a sound enamel area (S) and two separate, unprotected enamel windows for creating the artificial caries lesions. Specimens were stored in a demineralization solution [45] containing 50 mM acetic acid, 3 mM CaCl2 âˆ™ 2H2O, 3 mM KH2PO4 and 6 µl Methyl-hydroxy-diphosphonate (pH 4.95; 37 °C) with the enamel surfaces facing up for 22 d. The demineralization solution was prepared to cover the specimens excessively. pH was monitored daily and adjusted with 10 % hydrochloric acid and 10 M potassium hydroxide solution, if necessary.“

We only used enamel of permanent bovine teeth which is in accordance with the established, common standard in caries research. Since this approach is only designed to be applied in permanent teeth (scientifically as well as in potential clinical application) the discussion about demineralization of primary bovine teeth does not seem relevant to us.

5- Where is the sample size calculation?

We added a paragraph with the sample size calculation:

Materials and Methods (Subjection 2.1. Study design (1st paragraph)):

“An a priori power analysis was conducted using Sample size calculator (Version 1.061) for sample size estimation. Based on a previous study [26] the effect size was considered to be small using Cohen's (1988) criteria. With a significance level of α = .05 and power = .80, the minimum sample size required was calculated to be 15 assuming a drop-off rate of 10%.”

6- You repeat the conclusion twice at the end of discussion section and in the discussion section.

This has been corrected. The conclusion now is only in the Conclusion section.

7- Please make the limitations of the study.

The limitations of this study are described in the last paragraph of the discussion section. The discussion of limitations was expanded with two sentences about the bovine teeth model.

8- For the results put the standard deviation with the values as quantitative not only boxplot.

Standard deviation is only valuable to describe the dispersion in normal quantitative variables. As in our study the variable was not normally distributed, we used the interquartile range (Q25/Q75) to display the dispersion of data.

Reviewer 2 Report

Comments and Suggestions for Authors

Dear Authors,

the study is interesting and prepared and described with necessary detailes.

I have a few comments.

Does the study need an aprovement from the Local Bioethic Committee? I didn't notice any information about this.

Maybe also the scheme of samples treatment could be valuable for readers.

Section "Conclusions" is repeated. Please correct it.

Comments on the Quality of English Language

No comments, please check all spellings.

Author Response

We would like to thank the reviewers for the considerations and constructive feedback to help to improve the manuscript. Below please find our responses.

Reply to reviewer #2

1- Does the study need an aprovement from the Local Bioethic Committee? I didn't notice any information about this.

We added the necessary information in the back matter section IRBS (Institutional Review Board Statement):

Institutional Review Board Statement: The bovine incisors used in this study were obtained from a local slaughterhouse (VION, Bad Bramstedt, Germany, vionfoodgroup.com) after the cattle had been slaughtered for non-scientific reasons (meat production). Therefore, this study needed no ethical approval.

2- Maybe also the scheme of samples treatment could be valuable for readers.

In addition to figure 1, we provided a graphical abstract that depicts part of the treatment process of the specimens. We hope this will add to the understanding of the readers to the study design.

3- Section "Conclusions" is repeated. Please correct it.

This has been corrected (see above).

Round 2

Reviewer 1 Report

Comments and Suggestions for Authors

The paper was improved

Comments on the Quality of English Language

moderate

Author Response

Thank you for reviewing our paper. The manuscript was revised accordingly and has been linguistically revised, as well.